# High-Altitude Hypoxia Exposure Induces Iron Overload and Ferroptosis in Adipose Tissue

**DOI:** 10.3390/antiox11122367

**Published:** 2022-11-29

**Authors:** Yanfei Zhang, Jinyu Fang, Yingyue Dong, Huiru Ding, Quancheng Cheng, Huaicun Liu, Guoheng Xu, Weiguang Zhang

**Affiliations:** 1Department of Anatomy, School of Chinese Medicine, Beijing University of Chinese Medicine, Beijing 102488, China; 2Department of Anatomy and Embryology, Peking University School of Basic Medical Sciences, Beijing 100191, China; 3Department of Physiology and Pathophysiology, Peking University School of Basic Medical Sciences and Peking University Center for Obesity and Metabolic Disease Research, Beijing 100191, China

**Keywords:** high altitude, hypobaric hypoxia, adipose tissue, iron overload, ferrous iron, ferroptosis

## Abstract

High altitude (HA) has become one of the most challenging environments featuring hypobaric hypoxia, which seriously threatens public health, hence its gradual attraction of public attention over the past decade. The purpose of this study is to investigate the effect of HA hypoxia on iron levels, redox state, inflammation, and ferroptosis in adipose tissue. Here, 40 mice were randomly divided into two groups: the sea-level group and HA hypoxia group (altitude of 5000 m, treatment for 4 weeks). Total iron contents, ferrous iron contents, ROS generation, lipid peroxidation, the oxidative enzyme system, proinflammatory factor secretion, and ferroptosis-related biomarkers were examined, respectively. According to the results, HA exposure increases total iron and ferrous iron levels in both WAT and BAT. Meanwhile, ROS release, MDA, 4-HNE elevation, GSH depletion, as well as the decrease in SOD, CAT, and GSH-Px activities further evidenced a phenotype of redox imbalance in adipose tissue during HA exposure. Additionally, the secretion of inflammatory factors was also significantly enhanced in HA mice. Moreover, the remarkably changed expression of ferroptosis-related markers suggested that HA exposure increased ferroptosis sensitivity in adipose tissue. Overall, this study reveals that HA exposure is capable of inducing adipose tissue redox imbalance, inflammatory response, and ferroptosis, driven in part by changes in iron overload, which is expected to provide novel preventive targets for HA-related illness.

## 1. Introduction

Iron is known as an essential element, being indispensable for numerous metabolic processes, including oxygen delivery, electron transport, and enzymatic activity [1,2,3]. However, if in excess, iron also has potential toxicity, resulting in oxidative damage and lipid peroxidation via the Fenton reaction to produce reactive oxygen species (ROS) [2]. Therefore, iron is tightly regulated at the cellular and systemic levels to prevent deficiency and overload [3]. Hypobaric hypoxia is the most prominent feature of a high-altitude (HA) environment, in which the increase in hemoglobin synthesis and erythropoiesis requires more iron utilization [4,5], and adequate iron availability is an important prerequisite for adaptation to HA.

Iron homeostasis is vital to health and is maintained in a balanced state under normal conditions [6]. However, numerous studies revealed that HA hypoxia exposure induced significant changes in iron metabolism (uptake, storage, and efflux), causing iron homeostasis imbalance and even leading to toxicities such as oxidative stress [7,8,9], inflammatory response [10,11], and ferroptosis [12], an iron-dependent programmed cell death marked by ROS accumulation. As reported, chronic HA hypoxia causes brain iron accumulation and dysfunctional iron metabolism, inducing oxidative stress and apoptosis, and ultimately triggering cognitive impairment [7]. In addition, serum iron concentration, ferritin concentration, and transferrin saturation were all decreased after acute HA exposure, accompanied with an elevation in circulating proinflammatory cytokines, but a decrease in anti-inflammatory factors, leading to a systemic inflammatory response [10]. Moreover, HA exposure aggravated oxidative damage by producing MDA and inhibiting antioxidant enzyme activities in the liver, which could be reversed by iron supplementation with Fe-glycinate chelate (Fe-Gly) [8]. Better yet, acute HA exposure induced neuronal ferroptosis by iron elevation and formaldehyde accumulation, subsequently inducing neurological deficits [12].

Iron plays a critical role in the physiological functions and development of adipose tissue, and iron’s accumulation and reactivity were related to a range of adipose-related metabolic diseases [13,14,15]. Abnormal iron deposition is progressively considered a critical initiating factor of cell death, usually related to toxic free radicals and pathological damage [16]. In addition, high iron levels in adipose tissue can always put individuals at enhanced insulin resistance or diabetes risk [17]. Although the pathogenesis of iron deposition has been widely studied, research concerning the potential environmental conditions that may affect iron levels in adipose tissue is limited. To our knowledge, there are also no relevant studies on whether HA exposure affects iron homeostasis in adipose tissue. This study aims to better understand the effects of HA hypoxia on iron homeostasis, iron content, and ferroptosis in adipose tissue.

We assume that HA exposure may induce an imbalance in iron homeostasis in adipose tissue. Therefore, this study first detected whether HA could ignite iron overload in subcutaneous (scWAT), epididymal (eWAT), and interscapular brown (iBAT) adipose tissues. The contents of ROS release, 4-hydroxynonenal (4-HNE), malondialdehyde (MDA), glutathione (GSH), and the activities of catalase (CAT), superoxide dismutase (SOD), and glutathione peroxidase (GSH-Px), a series of indicators of oxidative damage and lipid peroxidation, were tested. In addition, proinflammatory factors in adipose tissue were assessed in response to HA exposure. Finally, the ferroptosis-related genes and relevant proteins were measured. Overall, this study revealed that HA exposure resulted in total iron and ferrous iron elevation, lipid peroxidation, redox imbalance, as well as inflammation, triggering ferroptosis susceptibility with enhanced ferroptotic events in adipose tissue. These results highlight the importance of the crosstalk among iron overload, redox reactions, inflammation, and ferroptosis in adipose tissue during HA exposure, hinting that adipose-tissue iron overload in HA environments might be a potential inducer triggering high-altitude illness and is expected to allow for the exploration of new preventive and therapeutic strategies for HA-related illness.

## 2. Materials and Methods

### 2.1. Animal and High-Altitude Model Establishment

All animal procedures were conducted in accordance with the standards of the NIH guidelines and the Institutional Animal Care and Utilization Committee of Peking University. All male C57BL/6 mice (18–20 g) were moved to and maintained in the experimental environment for adaptation to a 12-h light/dark cycle no less than 1 week before the experiment and were provided with free access to water and food in the Laboratory Animal Center at Peking University. Mice were randomized into two groups: the sea-level group and high-altitude hypoxia group. The high-altitude hypoxia group mice were placed in a hypobaric hypoxia chamber (ProOx-810, Shanghai Tawang Technology Co., Ltd., Shanghai, China); they ascended to a simulated altitude of 5000 m (HA, equivalent to 10.8% O_2_, 54.02 kPa) at 166 m per minute and descended to sea level at the same rate. The chamber altitude, oxygen and carbon dioxide levels, pressure, humidity, and temperature were continuously monitored. The chamber was opened to perform cage maintenance, cleaning and replenishing food and water, every 10 days. Mice in the sea-level control groups were housed in conventional cages but not within the HA chamber. After 4 weeks of treatment, all mice were sacrificed, and their subcutaneous white adipose tissue (scWAT), epididymal white adipose tissue (eWAT), and interscapular brown adipose tissue (iBAT) were separated and collected for further assessment.

### 2.2. Iron Assay

The total iron and ferrous iron levels in the adipose tissue were determined by the use of iron assay kits (Abcam, ab83366) in accordance with the relevant instructions. Briefly, adipose tissues (0.1 g) were added and homogenized with 1 mL of iron assay buffer. After full homogenization, adipose tissue homogenates were centrifuged to obtain the supernatant for detection and analysis at 16,000 rpm, 4 °C for 10 min. The absorbance at OD 570 nm was detected using 96 microplate readers. The levels of total nonheme iron and ferrous iron were calculated according to the manufacturer’s protocol.

### 2.3. Measurement of Reactive Oxygen Species (ROS)

The release of ROS was determined using an ROS assay kit (Dogesce, Beijing, China) according to the manufacturer’s protocols. In brief, 50 mg tissues were collected, separated, and homogenized with 500 μL of phosphate-buffered saline (PBS) buffer, and the supernatant was obtained by centrifugation. Next, the cell suspensions were centrifuged for 10 min at 500× *g*. Subsequently, the cell pellets were collected and washed three times with cooled PBS, and then resuspended at a density of 5 × 106 cells. The cell suspensions were coped with 0.1 mM 2′,7′-dichlorodihydrofluorescein diacetate (DCFH-DA) for 60 min at 37 °C. Finally, the fluorescence intensities were detected with an emission wavelength of 530 nm and an excitation wavelength of 490 nm.

### 2.4. Total RNA Extraction and Real-Time PCR 

Total RNA was isolated and extracted from the adipose tissue after treatment by the use of RNAtrip reagent (Applygen Technologies, Beijing, China) and following the standard instructions. Reverse transcription was conducted using the All-In-One RT MasterMix kit (Abm, Richmond, BC, Canada) in accordance with the manufacturer’s protocols. Subsequently, mRNA expression was detected via quantitative real-time RT−qPCR on a Stratagene Mx3000P system (Agilent Technologies, La Jolla, CA, USA) using RealStar Fast SYBR qPCR Mix (Genstar, Beijing, China). An 18S RNA was regarded as the internal control to normalize the value. The target gene expression levels were normalized to the 18S RNA using the comparative Ct method (2^−ΔΔCt^). The sequences of RT−qPCR primers are listed in Appendix A. 

### 2.5. Histomorphological Observation 

ScWAT, eWAT, and iBAT were separated and fixed in formalin buffer and used for histological examination. H&E was conducted according to the standard procedures. Perls-Diaminobenzidine (DAB) (Genmed, Shanghai, China) was performed using DAB substrate kits. Subsequently, the adipose tissue sections were observed using a microscope and quantitatively analyzed using ImageJ. At least ten views were selected randomly and adopted to perform the analysis.

### 2.6. Western Blotting

Adipose tissue samples were lysed with cold RIPA lysis buffer (0.5% Nonidet P-40, 0.1 M NaCl, 0.03 M HEPES with pH 7.6, protease inhibitors), and then the lysate was centrifuged at 13,000× *g* rpm, 4 °C for 15 min. The supernatants were collected and the concentration of the total proteins was measured using a BCA protein detection kit (Applygen Technologies, Beijing, China). Following this, 25 μg of protein were separated using SDS-PAGE gel and were moved onto NC membranes (Applygen Technologies, Beijing, China). Subsequently, the membranes were blocked with 2.5% nonfat milk and were incubated overnight with primary antibodies against anti-rabbit GPX4 (ABclonal, Wuhan, China), anti-rabbit SLC7A11 (ABclonal, Wuhan, China), anti-rabbit TFR1 (Invitrogen, Carlsbad, CA, USA), anti-rabbit L-ferritin (Abcam, Cambridge, UK), anti-rabbit H-ferritin (Abcam, Cambridge, UK), and anti-mouse β-actin (Cell Signaling Technology, Boston, MA, USA). After being washed three times, the membranes were treated with a corresponding horseradish peroxidase-conjugated secondary antibody (EASYBIO, Beijing, China). Finally, the protein bands were identified using an ECL kit (Applygen Technologies, Beijing, China). β-actin was served as the loading control.

### 2.7. Enzyme-Linked Immunosorbent Assay

The quantifications of 4-hydroxynonenal (4-HNE), malondialdehyde (MDA), glutathione (GSH), interleukin-6 (IL6), interleukin-1β (IL1β), and tumor necrosis factor α (Tnf-α) were analyzed using ELISA kits (Dogesce, Beijing, China). The enzyme activities of catalase (CAT), glutathione peroxidase (GSH-Px), and superoxide dismutase (SOD) were assayed using ELISA kits (Dogesce, Beijing, China). All assays were performed in triplicate.

### 2.8. Quantification and Statistical Analysis

Data are presented as the mean ± SEM. GraphPad Prism 6.0 was applied for the data analyses and statistical graph processing. Differences between the two samples were analyzed for statistical significance using the Student’s *t*-test. *p* values below 0.05 were considered statistically significant.

## 3. Results

### 3.1. Alterations in Hematological Parameters and Lipid Profiles following HA Exposure

HA exposure can induce a variety of adaptive or inadaptive physiological changes, such as changes in erythrocyte counts, hemoglobin concentrations, and lipid profiles [5]. Hematological analysis showed that mice had significantly higher RBC counts, higher hematocrit (HCT) levels, and higher mean corpuscular volume (MCV) levels after exposure to HA (*p* < 0.01, Figure 1a–c). In addition, the hemoglobin (HGB) concentration, mean corpuscular hemoglobin concentration (MCHC), and mean corpuscular hemoglobin (MCH) were also higher in mice at HA than mice in the sea-level group (*p* < 0.01 or *p* < 0.05, Figure 1d–f). For the biochemical measurements, total triglycerides and low-density lipoprotein (LDL) exhibited a remarkable increase, while total cholesterol and high-density lipoprotein (HDL) decreased markedly among HA mice when compared with the control group (*p* < 0.01, Figure 1g–j). Additionally, serum iron parameters were also altered remarkably (Appendix A). These results revealed that HA exposure could lead to a dramatic alteration in hematological parameters and lipid profiles.

### 3.2. HA-Exposure-Induced Iron Overload in Adipose Tissue 

Mice in the HA chamber demonstrated decreased adipose tissue weight compared with controls (*p* < 0.01 or *p* < 0.05, Figure 2a,b). Moreover, the appearance of subcutaneous (scWAT), epididymal (eWAT), and interscapular (iBAT) adipose tissue was much smaller and darker following HA exposure (*p* < 0.01, Figure 2a). To better explain the observed decrease in fat accumulation, we undertook histological analyses and showed that the size of the adipocytes of both scWAT and eWAT was significantly reduced in HA mice compared with the control group (*p* < 0.01, Figure 2c,d). Nevertheless, the volume of adipocytes in iBAT clearly increased (*p* < 0.01, Figure 2c,d). Next, we examined the effect of HA exposure on the iron content of adipose tissue using DAB-enhanced Perls’ staining and colorimetry and observed that iron was significantly deposited both in scWAT and BAT (Figure 2c). The eWAT had slight iron accumulation (Figure 2c). Simultaneously, colorimetry also showed a remarkable increase in total iron and ferrous iron content both in WAT and BAT (*p* < 0.01, Figure 2e). Taken together, these findings suggest that HA treatment reduced adipocyte hypertrophy and induced an increase in iron content, namely, iron overload in adipose tissue.

### 3.3. Lipid Peroxidation and Redox Imbalance of Adipose Tissue Caused by HA Exposure 

To test the effects of HA on the oxidant and antioxidant systems of the adipose tissue, ROS production, lipid peroxidation, and the alteration of the oxidative enzyme system were examined in three depots of adipose tissue, respectively. As shown in Figure 3, HA exposure induced a marked rise in the generation of ROS both in WAT and BAT (*p* < 0.01). MDA and 4-HNE are biomarkers and end products of lipid peroxidation [18,19]. Figure 4a,b reveals that the HA group showed strikingly higher MDA and 4-HNE levels than the control group (*p* < 0.01). SOD, CAT, and GSH-Px, three important antioxidant enzymes, are capable of decomposing toxic peroxides into nontoxic compounds [20], which reflect the potential of tissues or cells to resist oxidative stress. According to the results, the activities of SOD, CAT, and GSH-Px in the HA mice were considerably lower than those in the control group (*p* < 0.01, Figure 4c–e). Additionally, the tripeptide GSH is a major endogenous antioxidant for maintaining cellular redox homeostasis, which can protect the formation and function of the cytomembrane from superoxide [21]. As seen in Figure 4f, GSH levels were also decreased remarkably in response to HA exposure, which suggests that HA exposure markedly destroys cellular antioxidant capacity in both WAT and BAT. Combined with the index measured above, this suggests that HA hypoxia could cause oxidative damage and demolish the oxidant enzymes of adipose tissue.

### 3.4. Increased Secretion of Proinflammatory Factors in Adipose Tissue from HA Exposure

Subsequently, inflammatory factors induced by HA exposure were detected. According to the RT−qPCR results, most of the proinflammatory cytokines, including tumor necrosis factor α (Tnfα), interleukin-6 (IL6), interleukin-1β (IL1β), iNOS (inducible nitric oxide synthase), CD11c, and chemokine C-C motif chemokine ligand 2 (CCL2) were elevated in both WAT and BAT following HA treatment (*p* < 0.01 or *p* < 0.05, Figure 5a–c). The protein expression changes in Tnfα, IL6, and IL1β were shown in Figure 5d–f. As expected, their protein secretions (Figure 5d–f) in the HA mice were also significantly greater than those in the control. These results confirmed that HA exposure could induce an inflammatory response in adipose tissue.

### 3.5. HA Significantly Increased Ferroptosis Biomarkers in Adipose Tissue 

Ferroptosis is an iron-dependent oxidative cell death caused by oxidative stress and lipid peroxidation [22,23], which is usually accompanied by the dysfunction of these genes, such as the lipid repair enzyme glutathione peroxidase 4 (GPX4), solute carrier family 7 member 11 (SLC7A11), CHAC glutathione-specific gamma-glutamylcyclotransferase 1 (CHAC1), heme oxygenase-1 (HMOX1), as well as the iron-related genes transferrin receptor 1 (TFR1), H-ferritin (FTH), and L-ferritin (FTL). To test the effect of HA exposure on ferroptosis in adipose tissue, RT−qPCR and western blotting were used to measure the expression of ferroptosis-related biomarkers. The results indicated that GPX4, a major lipid-peroxidation scavenger and a key regulator of ferroptosis [24], was considerably downregulated at the mRNA and protein levels in scWAT and iBAT, but no significant changes were observed in eWAT in HA mice (*p* < 0.01, Figure 6). For the marker of SLC7A11, a critical factor of the cystine-glutamate antiporter inducing ferroptotic responses [25,26], it is differently regulated in three adipose tissue depots, with the upregulation at the mRNA and protein levels in eWAT and downregulation at the protein levels in scWAT and iBAT (*p* < 0.01 or *p* < 0.05, Figure 6). Meanwhile, CHAC1 and HMOX1, recognized markers of ferroptosis [27,28], were dramatically enhanced in the mRNA levels after HA exposure (*p* < 0.01 or *p* < 0.05, Figure 6). TFR1 and ferritin were detected to assess the intracellular iron uptake and storage abilities [29,30], and the observations indicated that there was a significant fall in TFR1 and FTH accompanied by a rise in FTL both in WAT and BAT (*p* < 0.01 or *p* < 0.05, Figure 6). Ferroportin-1 (Fpn1) is the only nonheme iron exporter found in mammals, and its loss can induce neuronal ferroptosis promoting memory impairment [31]. In the present study, we found Fpn1 mRNA and protein expression were significantly downregulated in scWAT, and there was slight and insignificant downregulation at the protein levels in eWAT and iBAT, although there was only a significant downregulation at the mRNA level (Appendix A). 

## 4. Discussion

Studies have shown that HA exposure elicits remarkable alterations in iron metabolism, leading to disturbances in iron homeostasis and giving rise to toxicities such as inflammation, oxidative stress, and ferroptosis [7,8,9,10,12]. Yet, as far as we know, no relevant study has examined these effects on adipose tissue during HA treatment. This report aims to examine the effect of HA hypoxia on iron metabolism, redox state, inflammatory response, and ferroptosis in adipose tissue. In this study, we demonstrated that HA exposure can promote iron homeostasis imbalance in WAT and BAT, which is related to iron overload, lipid peroxidation, redox imbalance, and inflammation, as well as ferroptosis. To our knowledge, this is the first report demonstrating that HA hypoxia induces iron overload and ferroptosis in adipose tissue.

Iron is an essential trace metal in almost all biological organisms [1,2]. However, iron contents need to be precisely regulated, as excess iron can exert toxicity by the Fenton reaction to produce powerful ROS, inducing oxidative damage [2]. Previous studies have shown that long-lasting HA hypoxia significantly increased brain iron concentrations, especially in deep gray matter regions, inducing cognitive impairment and neural injury [7,9]. We investigated the changes in iron levels by iron staining and colorimetry in adipose tissue as a result of HA exposure. The current research revealed, for the first time, that HA exposure induced iron overload in both WAT and BAT.

HA exposure is closely related to oxidative stress susceptibility [7,8,9]. An association between HA exposure and increased ROS generation has been demonstrated [32,33,34]. Moreover, the antioxidant systems of enzymes and nonenzymes were disturbed by HA exposure [34]. Uniformly, our results exhibit a significant rise in ROS production in both WAT and BAT of HA mice. Moreover, the levels of polyunsaturated fatty acid peroxidation end products MDA and 4-HNE, assessing the degree of oxidative stress, were increasingly higher than those of controls. Additionally, the activities of antioxidant enzymes SOD, CAT, and GSH-Px, comprising the first line of defense against oxidative damage [20], and the levels of GSH, a major endogenous antioxidant for maintaining cellular redox homeostasis [21], were markedly decreased. It should be emphasized that, under iron overload, excessive iron is a powerful driving force for oxidative stress as it catalyzes the reaction that produces highly toxic hydroxyl radicals. Combined with the above results, there is a good reason to believe that HA-mediated redox imbalance might be associated with an overload of adipose tissue.

Inflammation is the physiological defense of living organisms, but when the activation of inflammation is out of balance, it may cause various diseases [35]. Relevant studies have shown that HA plays a major role in the course of the systemic inflammatory response [10,11]. In addition, iron-dependent metabolic reprogramming has been shown to participate in the pathogenesis of inflammation [36]. In particular, ROS accumulation also helps to promote the inflammatory response and cause additional damage [37,38]. In our study, we found that HA exposure significantly elevated the expression of proinflammatory biomarkers, such as Tnfα, IL6, and IL1β, in both WAT and BAT. All these results indicated that the adipose tissue inflammatory response triggered by HA may be closely related to iron overload and iron-overload-induced ROS accumulation.

Ferroptosis is a novel kind of self-regulated cell death characterized by excessive iron-dependent oxidative stress and lipid peroxidation [22,23]. Acute exposure to HA may induce neuronal iron elevation and ferroptosis, subsequently leading to neurological deficits [12]. We investigated alterations in the ferroptosis signaling pathway in the adipose tissue of HA mice. According to the results, HA exposure induced ferroptosis, which was illustrated by a remarkable decline in the expression of GPX4, a key enzyme inhibiting ferroptotic damage [24], as well as a decrease in SLC7A11 in scWAT and iBAT. Intriguingly, HA exposure unexpectedly increased the protein levels of SLC7A11 in eWAT, triggering ferroptosis. One reasonable explanation is that eWAT initiates the negative feedback regulation mode against ferroptosis by increasing the expression of SLC7A11, owing to an excessive depletion of GSH in response to HA exposure. Similarly, Wang et al. reported that PM2.5 exposure induced ferroptosis in human endothelial cells by upregulating SLC7A11 [26]. Another study also showed that SLC7A11 may be involved in protecting iron-processed bone-marrow-derived macrophages from ferroptosis [39]. In addition, CHAC1 and HMOX1, ferroptosis-related genes capable of degrading glutathione and catalyzing the region-specific hydroxylation of heme to generate ferrous iron, respectively [27,28], were also enhanced upon exposure to HA. TFR1 is a major iron uptake protein, and a newly identified ferroptosis-related protein [29,40]. Unexpectedly, TFR1 is downregulated in the adipose tissue of HA mice, which can be explained as a compensation mechanism for limiting excessive iron uptake. Ferritin is an iron storage protein composed of 24 H-ferritin (FTH) and L-ferritin (FTL) subunits in ratios varying in different cell and tissue types [41]. In contrast to FTL, FTH possesses ferroxidase activity and converts iron from Fe^2+^ to Fe^3+^, the nontoxic form used for storage [42]. FTH suppression or deficiency increases sensitivity to ferroptosis, suggesting that ferritin may be involved in ferroptosis [30]. Our results show that the expression of FTH induced by HA was significantly reduced, while FTL markedly increased. According to the above data, it can be proved that there is obvious total iron and ferrous iron accumulation in adipose tissue after exposure to HA. The imbalance of iron uptake and storage mediated by these proteins accelerates the occurrence of ferroptosis.

Taken together, our results provide evidence that exposure to HA induces iron overload both in WAT and BAT, accompanied by lipid peroxidation, oxidative stress, and an inflammatory response. In addition, we revealed that ferroptosis can be induced by HA exposure, and the involved mechanisms are presented in the proposed scheme (Figure 7). The current study highlights that adipose tissue might be a promising therapeutic target for high-altitude-related illnesses and suggests an iron chelator as a protective measure. Further studies are warranted, including pharmacological interventions for HA-induced adipose iron overload, to investigate whether it can improve antioxidant and anti-inflammatory abilities.

## 5. Conclusions

In conclusion, our results demonstrate that HA exposure induced iron overload, lipid peroxidation, redox imbalance, and inflammation both in WAT and BAT. Moreover, we showed for the first time that HA hypoxia affects the expression of ferroptosis-related genes and proteins, leading to ferroptosis in adipose tissue. Notably, our findings hint that adipose-tissue iron overload during HA exposure might be a potential inducer triggering high-altitude illness, which is expected to allow for the development of novel preventive and therapeutic strategies targeting the regulation of iron homeostasis in adipose tissue.

## Figures and Tables

**Figure 1 antioxidants-11-02367-f001:**
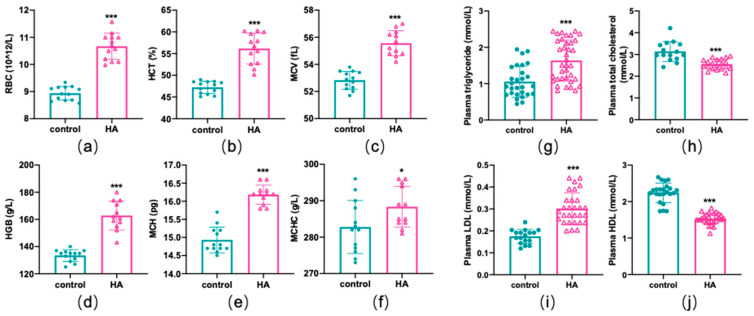
Hematological parameters and lipid profile differences induced by HA exposure. Graphs (**a**–**f**) display changes in hematological parameters. Graphs (**g**–**j**) display changes in lipid profiles. Data are presented as mean ± SEM of 12 mice. * *p* < 0.05 and *** *p* < 0.001 when compared with the control group.

**Figure 2 antioxidants-11-02367-f002:**
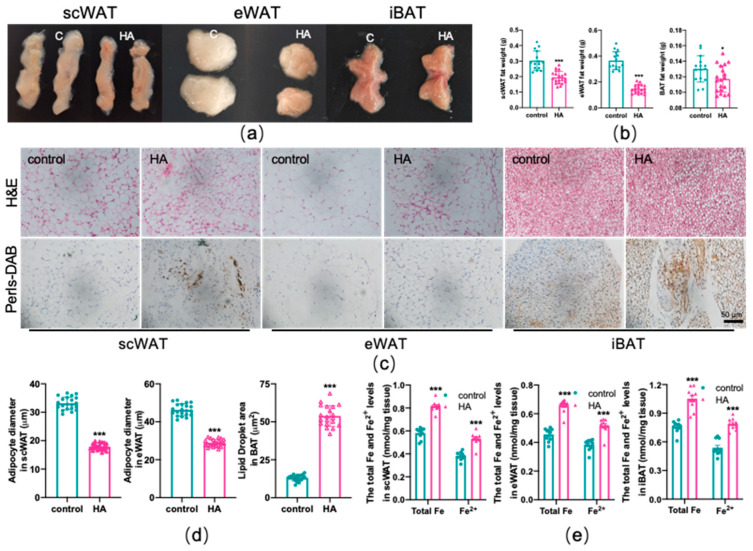
Reduced adipocyte hypertrophy and iron overload in adipose tissue due to HA exposure. (**a**) Representative images of scWAT, eWAT, and iBAT in mice. (**b**) Adipose tissue weight. (**c**) Representative images of fat pad sections stained with H&E and DAB enhanced Perls’ staining. (**d**) Diameter of adipocytes. (**e**) Total and ferrous iron content assay by colorimetry (*n* = 10). Labeled asterisks represent statistical significance in comparison with the control. * *p* < 0.05, and *** *p* < 0.001.

**Figure 3 antioxidants-11-02367-f003:**
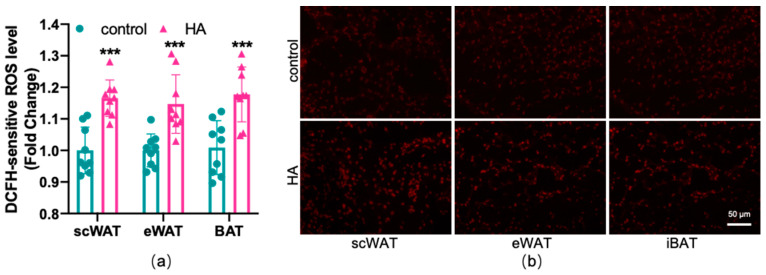
Effect of HA exposure on ROS generation in adipose tissue. (**a**) ROS generation determined by DCFH-DA (*n* = 9). (**b**) Representative images of fat pad sections stained with dihydroethidium (DHE) staining. The results are presented as the mean ± SEM of three independent experiments. *** *p* < 0.001, compared with the control group.

**Figure 4 antioxidants-11-02367-f004:**
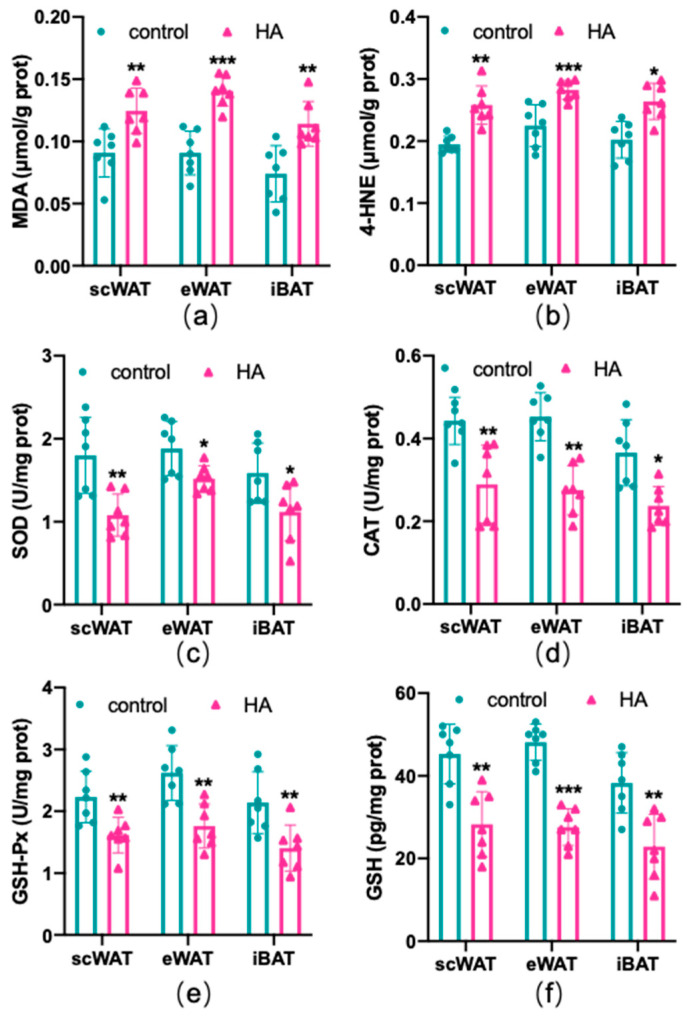
Oxidative stress-related factors in both WAT and BAT from HA exposure for 4 weeks. Graphs (**a**,**b**) display the levels of MDA and 4-HNE (ELISA). Graphs (**c**–**e**) display the enzyme activity of SOD, CAT, and GSH-Px (enzyme activity detection). Graph (**f**) displays changes in the expression of GSH (ELISA). Data are expressed as the mean ± SEM; *n* = 7. * *p* < 0.05, ** *p* < 0.01, and *** *p* < 0.001, compared with the control group.

**Figure 5 antioxidants-11-02367-f005:**
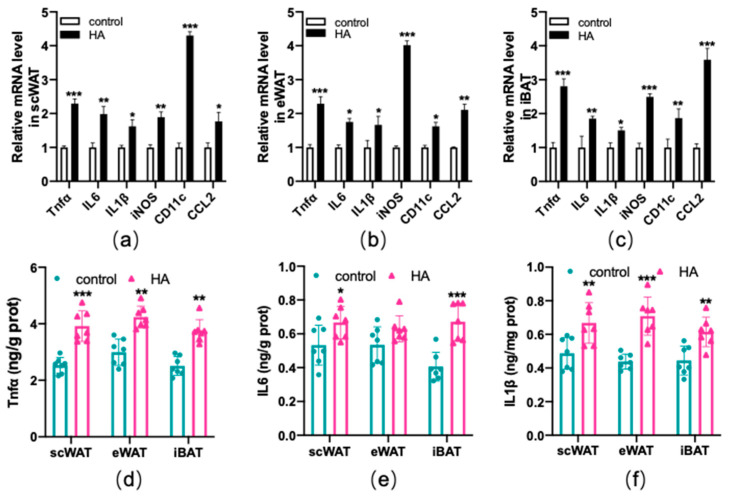
Production of inflammatory-related factors in both WAT and BAT after exposure to HA. Graphs (**a**–**c**) display RT−qPCR of inflammatory-related genes after HA exposure. Graphs (**d**–**f**) display Tnfα, IL6, and IL1β assay after 4-week-long exposure to HA (ELISA). Data are expressed as the mean ± SEM; *n* = 7. * *p* < 0.05, ** *p* < 0.01, and *** *p* < 0.001, compared with the control group.

**Figure 6 antioxidants-11-02367-f006:**
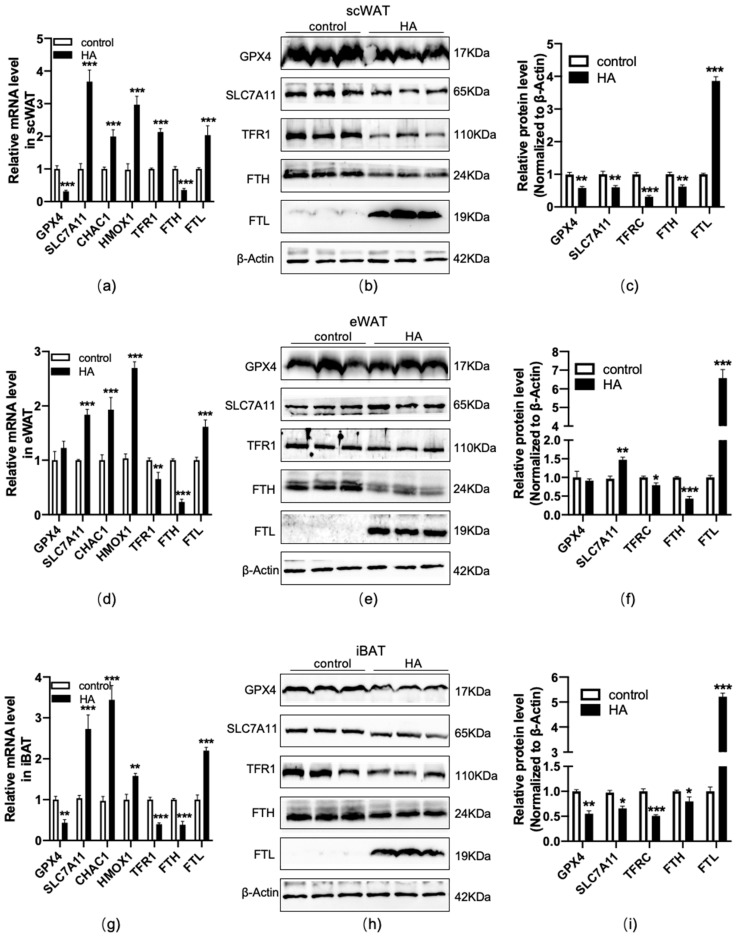
Changes in ferroptosis-related genes and proteins in adipose tissue during HA exposure. Graphs (**a**,**d**,**g**) display RT−qPCR of ferroptosis-linked genes and iron metabolism-related genes after HA exposure. Data are expressed as the mean ± SEM; *n* = 6. Graphs (**b**,**e**,**h**) display representative bands of ferroptosis-related protein levels; *n* = 3. Graphs (**c**,**f**,**i**) display quantification of the relative protein levels, normalized to β-actin. * *p* < 0.05, ** *p* < 0.01, and *** *p* < 0.001, compared with the control group.

**Figure 7 antioxidants-11-02367-f007:**
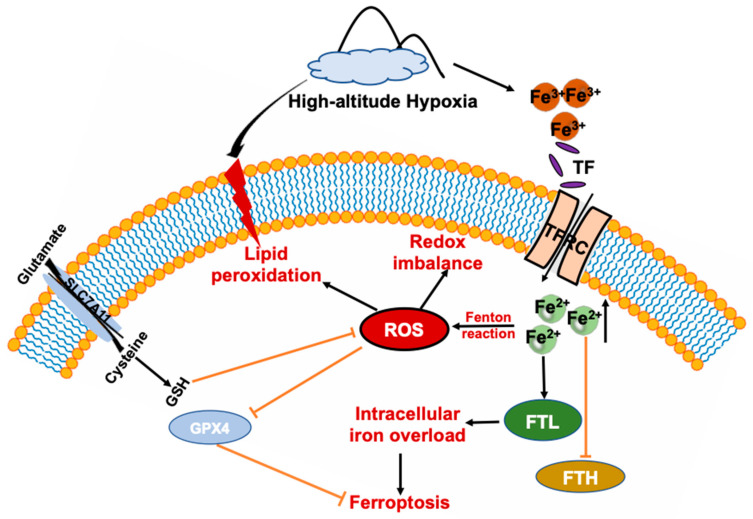
Schematic summary of HA-exposure-induced adipose tissue ferroptosis. HA disturbs iron homeostasis via differentially regulated iron uptake and storage-related genes and proteins such as TFR, FTH, and FTL, leading to iron overload in both WAT and BAT, that soon afterwards induces adipose tissue to produce massive amounts of ROS and cause redox imbalance, and lipid peroxidation. Ultimately, adipose tissue is more prone to ferroptosis response after exposure to HA.

## Data Availability

The data that support the findings of this study are available from the corresponding author upon reasonable request.

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
