# Peer review of "High-Altitude Hypoxia Exposure Induces Iron Overload and Ferroptosis in Adipose Tissue"

_antioxidants, 2022, doi:10.3390/antiox11122367_

Round 1

Reviewer 1 Report

The explanation under the charts needs to be slightly improved. I suggest: Graphes (a-f)  - changes in hematological parameters instead of: (a-f)  Changes in hematological parameters.

In the references section (3, 9, 13, 19, 22, 24, 27), some materials are missing page numbers.

The text of the work mentions an additional figure No. 1, but in the following part there is no additional figure 1

Author Response

  1. The explanation under the charts needs to be slightly improved. I suggest: Graphes (a-f) changes in hematological parameters instead of: (a-f) Changes in hematological parameters.

Response:

Thanks for reviewer’s suggestions. We have improved the descriptions of the figure legend accordingly.

  1. In the references section (3, 9, 13, 19, 22, 24, 27), some materials are missing page numbers.

Response:

Thanks for reviewer’s suggestions. We have corrected this information in the reference.

  1. The text of the work mentions an additional figure No. 1, but in the following part there is no additional figure 1.

Response:

The additional figure 1-2 were showed in supplementary data.

Reviewer 2 Report

The authors established a straight-forward experimental group by inducing high-altitude hypoxia after transporting the animals to an area of 500 meters above sea level. The authors found that the HA group was more exposed to oxidative stress by extracting adipose tissue from the control and experimental groups and measuring iron levels, ROS levels, and oxidative-related factors. Next, it was confirmed that the adipose tissue of the HA group expressed more inflammatory factors than the control group. By observing changes in ferroptosis-related genes and proteins, it was finally confirmed that HA induces ferroptosis in adipose tissue.

The authors' experimental data are immaculate and reliable, but since there is no data on the mechanism that connects each data, it is a giant leap to claim a schematic diagram like figure 7. Based on the results of this manuscript, the schematic image of figure 7 cannot be claimed.For example, the authors confirmed an increase in oxidative stress-related factors and an increase in inflammatory-related factors in the adipose tissue of the HA group. However, it cannot be argued that ROS increased the inflammatory factors of the HA group. Mechanism data is required to know the correlation between the two outcomes.

For an antioxidant journal accepts this manuscript, discussing the reason for the results and studying the deeper mechanism is necessary.

Author Response

The authors established a straight-forward experimental group by inducing high-altitude hypoxia after transporting the animals to an area of 500 meters above sea level. The authors found that the HA group was more exposed to oxidative stress by extracting adipose tissue from the control and experimental groups and measuring iron levels, ROS levels, and oxidative-related factors. Next, it was confirmed that the adipose tissue of the HA group expressed more inflammatory factors than the control group. By observing changes in ferroptosis-related genes and proteins, it was finally confirmed that HA induces ferroptosis in adipose tissue.

The authors' experimental data are immaculate and reliable, but since there is no data on the mechanism that connects each data, it is a giant leap to claim a schematic diagram like figure 7. Based on the results of this manuscript, the schematic image of figure 7 cannot be claimed. For example, the authors confirmed an increase in oxidative stress-related factors and an increase in inflammatory-related factors in the adipose tissue of the HA group. However, it cannot be argued that ROS increased the inflammatory factors of the HA group. Mechanism data is required to know the correlation between the two outcomes.

For an antioxidant journal accepts this manuscript, discussing the reason for the results and studying the deeper mechanism is necessary.

Response:

Thanks for reviewer’s suggestions. We have corrected the schematic diagram and descriptions of the figure 7 in the revised manuscript.

In addition, we discussed that the adipose tissue inflammatory response triggered by HA may be closely related to iron overload and iron overload-induced ROS accumulation in the fourth paragraph of the discussion: Inflammation is the physiological defense of living organisms, but when the activation of inflammation is out of balance, it may cause various diseases [35]. Relevant studies showed that HA plays a major role in the course of the systemic inflammatory response [10, 11]. In addition, iron-dependent metabolic reprogramming has been shown to participate in the pathogenesis of inflammation [36]. In particular, ROS accumulation also helps to promote the inflammatory response and cause additional damage [37, 38]. In our study, we found that HA exposure significantly elevated the expression of proinflammatory biomarkers such as Tnfα, IL6, and IL1β in both WAT and BAT. All these results indicated that the adipose tissue inflammatory response triggered by HA may be closely related to iron overload and iron overload-induced ROS accumulation.

Reviewer 3 Report

High-altitude environment is known to cause hypobaric hypoxia which may have deleterious effects on health. Hypobaric hypoxia is a form of stress that would increase haemoglobin synthesis and erythropoiesis to ensure adequate iron utilization of the body. Adipose tissue is increasingly recognised for its active endocrine role, beyond its lipid storage function. Moreover, adipose tissue has been shown to be sensitive to hypoxia-induced adaptation. The present study sough to investigate the effect of high-altitude related hypobaric hypoxia on adipose tissue, with a particular focus on iron homeostasis. The authors reported that high-altitude related hypobaric hypoxia increased total iron and ferrous iron 19 in both white and brown adipose tissues. This was accompanied by increased ROS release, MDA and 4-HNE elevation, GSH depletion, and diminished cellular antioxidant defence evidenced by reduced SOD, CAT, and GSH-Px activities, leading to a proinflammatory profile in the adipose tissue. The disrupted redox haemostasis and proinflammatory profile may be explained by alteration in ferroptosis signalling. The authors concluded that high-altitude related hypobaric hypoxia perturbs adipose tissue redox homeostasis, pro-inflammatory response, and ferroptosis, driven in part by changes in iron overload.

Concerns:

1.       Regarding high-altitude hypoxia, 5,000 meters was described (line 95) in the method without any details on how this was achieved in the study. This is important as the entire study is based on high-altitude hypoxia. If space is an issue, please include the relevant description and/or references in the supplementary.

2.       Figure 6, were the qPCR and western blot data from n = 6 or more? The method section described “Forty mice were randomized into two groups” implying n = 20. However, figure legend describes “The values are presented by three independent experiments”. Meanwhile, original image from the upload contains n= 3-6. Also line 159-162 of the methods, mentioned “ data are presented as the mean + SEM of at least 3 independent experiments”, was the animal experiment performed in 3 batches to gather n = 20? Please check and clarify how many were included into the respective analyses and update the descriptions accordingly.

3.       The diverged SLC7A11-mediated ferroptosis in the different adipose tissue should also be discussed.

4.       Line 155-157, were biochemical assays or ELISA kits used to measure enzyme activities?

5.       Figure 2, lipids are removed by the chemical solvents (e.g. histolene) under standard histological preparation, how did the authors ascertain lipid droplet area in BAT?

Specific comments:

1.       Line 124, “Reversing the separation…” to “Reverse transcription…”.

2.       Please add the analysis method for the qPCR, i.e. 2-delta-delta CT method.

3.       Line 131, the different adipose tissues abbreviations were already defined in the introduction.

4.       Line 151, “loading control” may be more appropriate.

5.       Suggest to change “±SEM” to “+SEM”.

6.       Line 164 and 239, change “during” to “following”.

7.       Line 237, please be consistent with the format use for the pro-inflammatory cytokines. If gene naming format is used, please present them in “Tnfα”.

8.       Line 242, change to “pro-inflammatory”.

9.       Line 264, space between “TFR1” and “and”.

10.   Line 288, change to “…produce massive amount of ROS…“.

11.   Line 327, delete the extra “over”.

Author Response

High-altitude environment is known to cause hypobaric hypoxia which may have deleterious effects on health. Hypobaric hypoxia is a form of stress that would increase haemoglobin synthesis and erythropoiesis to ensure adequate iron utilization of the body. Adipose tissue is increasingly recognised for its active endocrine role, beyond its lipid storage function. Moreover, adipose tissue has been shown to be sensitive to hypoxia-induced adaptation. The present study sough to investigate the effect of high-altitude related hypobaric hypoxia on adipose tissue, with a particular focus on iron homeostasis. The authors reported that high-altitude related hypobaric hypoxia increased total iron and ferrous iron 19 in both white and brown adipose tissues. This was accompanied by increased ROS release, MDA and 4-HNE elevation, GSH depletion, and diminished cellular antioxidant defence evidenced by reduced SOD, CAT, and GSH-Px activities, leading to a proinflammatory profile in the adipose tissue. The disrupted redox haemostasis and proinflammatory profile may be explained by alteration in ferroptosis signalling. The authors concluded that high-altitude related hypobaric hypoxia perturbs adipose tissue redox homeostasis, pro-inflammatory response, and ferroptosis, driven in part by changes in iron overload.

Concerns:

  1. Regarding high-altitude hypoxia, 5,000 meters was described (line 95) in the method without any details on how this was achieved in the study. This is important as the entire study is based on high-altitude hypoxia. If space is an issue, please include the relevant description and/or references in the supplementary.

Response:

Thanks for reviewer’s suggestions. We have improved the descriptions in the revised manuscript as below:

Mice were randomized into two groups: the sea-level group and high-altitude hypoxia group. High-altitude hypoxia group mice were placed in a hypobaric hypoxia chamber (ProOx-810, Shanghai Tawang Technology Co., Ltd), and proceeded to a simulated altitude of 5000 m (HA, equivalent to 10.8% O2, 54.02 kPa) at 166 m per minute and descent to sea level at the same rate. The chamber altitude, oxygen and carbon dioxide levels, pressure, humidity and temperature were continuously monitored. The chamber was opened to perform cage maintenance to clean and replenish food and water every 10 days. Mice in the sea-level control groups were housed in conventional cages but not within the HA chamber.

  1. Figure 6, were the qPCR and western blot data from n = 6 or more? The method section described “Forty mice were randomized into two groups” implying n = 20. However, figure legend describes “The values are presented by three independent experiments”. Meanwhile, original image from the upload contains n= 3-6. Also line 159-162 of the methods, mentioned “ data are presented as the mean + SEM of at least 3 independent experiments”, was the animal experiment performed in 3 batches to gather n = 20? Please check and clarify how many were included into the respective analyses and update the descriptions accordingly.

Response:

Thanks for reviewer’s suggestions. We have improved the descriptions in appropriate figure legend and method section accordingly.

  1. The diverged SLC7A11-mediated ferroptosis in the different adipose tissue should also be discussed.

Response:

Thanks for reviewer’s suggestions. SLC7A11, a critical factor of the cystine-glutamate antiporter inducing ferroptotic responses, it is differently regulated in three adipose tissue depots, with the upregulation at the mRNA and protein levels in eWAT and down-regulation at the protein levels in scWAT and iBAT (p < 0.01 or p < 0.05, Figure 6).

The discussion about it is in the fifth paragraph of the discussion: HA exposure induced ferroptosis, which was illustrated by a decrease in SLC7A11 in scWAT and iBAT. Intriguingly, HA exposure unexpectedly increased the protein levels of SLC7A11 in eWAT, triggering ferroptosis. One reasonable explanation is that eWAT initiates the negative feedback regulation mode against ferroptosis by increasing the expression of SLC7A11, owing to an excessive depletion of GSH in response to HA exposure. Similarly, Wang et al. reported that PM2.5 exposure induced ferroptosis in human endothelial cells by upregulating SLC7A11 [26]. Another study also showed that SLC7A11 may be involved in protecting iron-processed bone marrow-derived macrophages from ferroptosis in comparison to the control group [39].

  1. Line 155-157, were biochemical assays or ELISA kits used to measure enzyme activities?

Response:

Thanks for reviewer’s suggestions. The activities of superoxide dismutase (SOD), catalase (CAT), and glutathione peroxidase (GSH-Px) were tested by ELISA kits.

  1. Figure 2, lipids are removed by the chemical solvents (e.g. histolene) under standard histological preparation, how did the authors ascertain lipid droplet area in BAT?

Response:

Thanks for reviewer’s question. We showed the lipid droplet area in Figure 2d measured by NIH ImageJ program. In our opinion, although the lipids in the lipid droplets were removed during slice preparation, the vacuoles maintained their original shape in fixation, and the area of the lipid droplets can be characterized by measuring the area of the vacuoles. The area (μm2) or diameter (μm) of vacuoles were measured using NIH ImageJ program. Specifically, Fat pad was fixed in 4% phosphate-buffered formalin, dehydrated in ethanol and xylene, embedded in paraffin, cut into 6 μm sections, and stained with hematoxylin and eosin for microscopic examination. Four HE fields of each mouse were selected and 150 lipid droplets of each field were analyzed to measure vacuoles areas by NIH ImageJ program, in which the average value of 20 fields were analyzed for mapping.

Specific comments:

  1. Line 124, “Reversing the separation…” to “Reverse transcription…”.

Response:

Thanks. It was corrected.

  1. Please add the analysis method for the qPCR, i.e. 2-delta-delta CT method.

Response:

Thanks for reviewer’s suggestions. We have added the analysis method for the qPCR in the method of the revised manuscript.

  1. Line 131, the different adipose tissues abbreviations were already defined in the introduction.

Response:

Thanks for reviewer’s suggestions. we have added this information in the introduction accordingly.

  1. Line 151, “loading control” may be more appropriate.

Response:

Thanks. It was corrected.

  1. Suggest to change “±SEM” to “+SEM”.

Response:

Thanks. It was corrected.

  1. Line 164 and 239, change “during” to “following”.

Response:

Thanks. It was corrected.

  1. Line 237, please be consistent with the format use for the pro-inflammatory cytokines. If gene naming format is used, please present them in “Tnfα”.

Response:

Thanks. It was corrected.

  1. Line 242, change to “pro-inflammatory”.

Response:

Thanks. It was corrected.

  1. Line 264, space between “TFR1” and “and”.

Response:

Thanks. It was corrected.

  1. Line 288, change to “…produce massive amount of ROS…“.

Response:

Thanks. It was corrected.

  1. Line 327, delete the extra “over”.

Response:

Thanks. It was corrected.

Round 2

Reviewer 2 Report

Strictly speaking, Figure 7 still has many problems because there is no mechanism data. As this manuscript claims, the data on the signal changes of adipocytes caused by HA are straightforward and non-controversial. In other words, I am not saying that this paper has logical errors but is incomplete. Adding data that studied the mechanism will make the story even more beautiful, but since I respect the authors' efforts so far, I want to make a favorable judgment unlike before.

Author Response

We appreciate the reviewer’s insightful comments and suggestions. In the present study, we identify that HA exposure is capable of inducing adipose tissue redox imbalance, lipid peroxidation, and ferroptosis, driven in part by changes in iron overload, and it is enough to draw a qualitative conclusion that high-altitude hypoxia exposure induces iron overload and ferroptosis in adipose tissue. We agree with the reviewer that further research with more quantitative analysis will be a promising research direction, and we will do this as our future work.